# Virulence Variation of *Salmonella* Gallinarum Isolates through SpvB by CRISPR Sequence Subtyping, 2014 to 2018

**DOI:** 10.3390/ani10122346

**Published:** 2020-12-09

**Authors:** Koeun Kim, Sunghyun Yoon, Yeong Bin Kim, Young Ju Lee

**Affiliations:** College of Veterinary Medicine & Zoonoses Research Institute, Kyungpook National University, Daegu 41566, Korea; kke02062@gmail.com (K.K.); sungyoon@knu.ac.kr (S.Y.); kimybins@gmail.com (Y.B.K.)

**Keywords:** *Salmonella* Gallinarum, fowl typhoid, virulence, CRISPR

## Abstract

**Simple Summary:**

*Salmonella* Gallinarum causes fowl typhoid in all ages of chickens, which results in economic loss of commercial chicken farms. The disease has been eradicated in many developed countries, but is still prevalent in Korea. In this study, we investigated virulence and genetic variation of *S.* Gallinarum from Korea, between 2014 and 2018. The results indicated that virulence was increased, which was associated with genetic change over time. Therefore, surveillance of genetic change associated with virulence increase is necessary for monitoring of *S.* Gallinarum isolates for dissemination.

**Abstract:**

*Salmonella* Gallinarum is a Gram-negative bacteria that causes fowl typhoid, a septicemic disease with high morbidity and mortality that affects all ages of chickens. Although vaccines and antimicrobials have been used nationwide to eradicate the disease, the malady is still prevalent in Korea. In this study, we investigated the virulence and genetic variation of 116 *S*. Gallinarum isolates from laying hens between 2014 and 2018. A total of 116 isolates were divided into five Gallinarum Sequence Types (GST) through clustered regularly interspaced short palindromic repeats (CRISPR) subtyping method. The GSTs displayed changes over time. The 116 isolates showed no difference in virulence gene distribution, but the polyproline linker (PPL) length of the SpvB, one of the virulence factors of *Salmonella* spp., served as an indicator of *S*. Gallinarum pathogenicity. The most prevalent PPL length was 22 prolines (37.9%). The shortest PPL length (19 prolines) was found only in isolates from 2014 and 2015. However, the longest PPL length of 24 prolines appeared in 2018. This study indicates that PPLs of *S.* Gallinarum in Korea tend to lengthen over time, so the pathogenic potency of the bacteria is increasing. Moreover, the transition of GST was associated with PPL length extension over time. These results indicate that surveillance of changing GST and PPL length are necessary in the monitoring of *S*. Gallinarum isolates.

## 1. Introduction

Fowl typhoid (FT), caused by *Salmonella enterica* subsp. *enterica* serovar Gallinarum biovar Gallinarum (*S*. Gallinarum), is a host-adapted septicemic bacterial disease associated with high morbidity and mortality in all ages of poultry, and the infection of chicken has been a serious problem in Korea [1,2]. The disease has several transmission routes: through food, environment, or contaminated eggs, causing significant financial losses for farming industries. [1,3]. Therefore, FT critically affects chicken farms economically worldwide, except for those countries that are not suffering from the disease [3]. FT has been eradicated in many developed countries, including the United States [4], Canada, and most countries in Western Europe [1]. Although the policy for control and eradication of FT in Korea (similar to the National Poultry Improvement Plan in the United States) has been underway since the 1970s [5], the disease is still prevalent. Most of the commercial layer population consists of brown-shell egg layers, which are more susceptible to FT [2]; additionally, red poultry mites, which is a significant mechanical vector for transmission of FT, have spread to poultry houses nationwide [6]. Moreover, a live *S*. Gallinarum 9R strain vaccination was administered nation-wide to commercial layers since 2001, but recently, it has not provided sufficient protection against FT [7]. That means, recent field strains seem to be different from the vaccine strain, and, considering the prevalence of the disease, the bacteria must have experienced some evolution. Therefore, the present study aimed to determine whether the virulence of *S*. Gallinarum field strains have increased by molecular research.

The virulence of the bacteria is determined by virulence factors, which consist of virulence plasmids, toxins, fimbriae, flagellae, and genes of Salmonella pathogenicity islands [8]. In particular, the Salmonella plasmid virulence (spv) locus in virulence plasmids is necessary for the intracellular survival and replication of *Salmonella* spp. in macrophages, and the *spvB* of the locus synthesizes ADP-ribosylating toxins that destabilize cytoskeletons in host cells, playing a critical role in pathogenesis of *Salmonella* spp. [9,10]. Kwon and Cho [11] reported that a polyproline linker (PPL) connecting the N- and C-terminal domains of SpvB determines the pathogenic activity of SpvB, and Kim et al. [12] also reported that PPL length could be an indicator of the molecular evolution of *S*. Gallinarum pathogenicity.

A clustered regularly interspaced short palindromic repeats (CRISPR) locus consists of direct repeated sequences with highly variable spacers between the repeats and leader sequence, and it functions as a defense system against foreign invaders with CRISPR-associated genes in prokaryotes [13,14,15]. However, CRISPR regions are a significantly useful, economic, and effective method of subtyping [16]. This CRISPR-based subtyping method uses the polymorphism in the number and type of spacers [17] and shows good correspondence with established typing methods such as pulsed-field gel electrophoresis and whole genome sequencing typing [18,19]. Moreover, the method can separate genetically highly homogenous strain, while other subtyping methods are less efficient at stratifying the strain into several subgroups [19].

In this study, we investigated genetic variation by CRISPR-based typing and PPL lengths of SpvB as a virulence indicator, in *S*. Gallinarum isolates from 2014 to 2018, also the molecular evolution with regard to the virulence of *S*. Gallinarum field isolates was analyzed.

## 2. Materials and Methods

### 2.1. Bacterial Strains

A total of 116 *S.* Gallinarum isolates were isolated from post-mortem clinical cases of FT-affected chickens from 8 of 9 different provinces in Korea (Gyeonggi, Gangwon, Chungbuk, Chungnam, Jeonbuk, Jeonnam, Gyeongbuk, and Gyeongnam) as previously described [20]. Bacterial isolates were collected from 2014 to 2018 as follows: 37 in 2014, 13 in 2015, 15 in 2016, 23 in 2017, and 28 in 2018.

### 2.2. DNA Extraction

All *S*. Gallinarum isolates were grown on tryptic soy agar (BD, Sparks, MD, USA) at 37 °C for 24 h. After incubation, bacterial DNA was extracted with an AccuPrep Genomic DNA Extraction Kit (Bioneer, Daejeon, Korea) according to the manufacturer’s instructions.

### 2.3. Detection of Virulence Genes

We performed multiplex polymerase chain reaction (PCR) using the Accupower PCR PreMix (Bioneer) for identifying 20 virulence genes of *Salmonella* spp. using six reactions. The primers used in this study are listed in Table 1. Four of the six reactions were used for amplifying *spiA, cdtB,* and *msgA* (set 1), *invA, prgH, spaN, orgA,* and *tolC* (set 2), *sitC, lpfC, sifA,* and *sopB* (set 3), and *iroN* and *pagC* (set 4) under the following conditions: 5 min at 95 °C, 25 cycles of 30 s at 94 °C, 30 s at 66.5 °C, and 2 min at 72 °C, with a final cycle of 10 min at 72 °C. The remaining two reactions, which amplified *stn, sopE,* and *pefA* (set 5), and *sipB, sefC,* and *spvB* (set 6), were processed under the following conditions: 5 min at 94 °C, 35 cycles of 30 s at 94 °C, 1 min at 58 °C, 2 min at 72 °C, and a final cycle of 10 min at 72 °C. After PCR amplification, all products were analyzed with gel electrophoresis in 1.5% agarose gel with 100 bp Plus DNA ladder (Solgent, Daegeon, Korea).

### 2.4. Detection of CRISPR Loci and PPLs

CRISPR loci and PPL regions of all isolates were amplified by PCR and analyzed by sequencing as previously described [12,19]. Two CRISPR loci of *S*. Gallinarum were confirmed by CRISPRdb (https://crispr.i2bc.paris-saclay.fr/) [23]. Primers for amplification of two CRISPR loci, CRISPR1 and CRISPR2, which were verified with Basic Local Alignment Search Tool (BLAST) from the National Center for Biotechnology Information (http://www.ncbi.nlm.nih.gov/BLAST), are listed in Table 1. PCR products were purified with a QIAquick PCR purification kit (Qiagen, Hilden, Germany), and sequencing was conducted using an automatic sequencer (Cosmogenetech, Seoul, Korea). For sequence analyses, only spacers were extracted from each CRISPR sequence using the webtool “Institute Pasteur CRISPR database for Salmonella” (https://galaxy.pasteur.fr/?tool_id=toolshed.pasteur.fr%2Frepos%2Fkhillion%2Fsalmonella_crispr_typing%2Fsalmonella_crispr_typing%2F1.0.1&version=1.0.1&__identifer=7rjdf0q97u7) (ver. 1.0.1), as described by Fabre et al. [24]. Repeats were excluded from the analysis. GSTs were defined by the spacer combination of allelic types at the CRISPR1 and CRISPR2 loci, and a specific number was given to each type. PPL sequences were translated and analyzed using the SnapGene program (ver. 5. 0. 8) to identify PPL length of SpvB.

### 2.5. Statistical Analysis

We analyzed the data using SPSS 25 statistical software (IBM Corp., Armonk, NY, USA). Differences associated with PPL length, GSTs, and year were confirmed using Pearson’s chi-squared test. For cells with less than five factors, we used Fisher’s exact test for a more accurate estimate. Statistically significant differences (*p* < 0.05) were verified using post hoc Bonferroni correction, as previously described [25].

## 3. Results

### 3.1. Distribution of Virulence Genes

Among the 20 virulence genes analyzed, 18 genes were found in all isolates, whereas 2 genes (*cdtB* and *pefA*) were not found in any of the tested isolates. There was no difference in virulence gene distribution among all isolates from 2014 to 2018. Distribution of virulence genes of the 116 *S*. Gallinarum isolates are shown in Table 2.

### 3.2. Genetic Variation by CRISPR Sequence Typing

Spacer contents and arrangements of each CRISPR locus are shown in Figure 1. The names of spacers are listed under each allele, and they are named as Fabre et al. [24] described. CRISPR1 and CRISPR2 loci showed 3 and 5 CRISPR allelic types, respectively. Although the allelic types in locus 1 only had 2 spacers (except GST 4, which had 8 spacers), the 5 allelic types in locus 2 played an important role in dividing GSTs; all of them had different spacers. Of the 116 *S.* Gallinarum isolates, the most common type was GST 5 (81.9%, *n* = 95 isolates), followed by GST 2 (11.2%, *n* = 13 isolates), GST 3 (3.4%, *n* = 4 isolates), GST 4 (2.6%, *n* = 3 isolates), and GST 1 (0.9%, *n* = 1 isolate). GST 2 possessed the fewest spacers (7), while GST 4 possessed the most spacers (18). GST 1 had two novel spacers, which were not present in “Institute Pasteur CRISPR database for Salmonella” program database, so they are labeled as ‘A’ and ‘B’ in Figure 1.

GST distributions per each year from 2014 to 2018 are presented in Figure 2. GST 5 was found in isolates from all years with high rates of occurrence (67.6%–100%), but existed especially statistically significantly in 2014 and 2017. Alternatively, GST 2 existed in 2014, but showed rapid decrease from 2014 to 2016 (29.7% to 6.7%) and disappeared altogether after 2016. GST 1 only existed in isolates from 2014, while GST 3 appeared in 2016 and 2018, and GST 4 first emerged in 2018.

### 3.3. Variation of PPL Length of SpvB

Graphs reflecting PPL lengths are shown in Figure 3, indicating that the PPL lengths of *S*. Gallinarum isolates varied from 19 to 24 prolines. The most prevalent PPL length was 22 prolines (37.9%), followed by 21, 23, 20, 19, and 24 prolines (36.2%, 15.5%, 6.9%, 2.6%, and 0.9%, respectively). When compared by year, ranges of PPL lengths in 2016, 2017, and 2018 were 21–23, 20–23, and 21–24 prolines, respectively. The shortest PPL length of 19 prolines was found only in 2014 and 2015, whereas the longest PPL length of 24 prolines appeared in 2018. Also, 24 prolines showed significant difference (*p* < 0.05) according to year.

In comparison by GST, although GST 5 (which contained the most isolates from 2014 to 2018) had a wide range of PPL lengths (19–24 prolines), GST 1 (which included only 1 isolate from 2014) had a PPL length of 20 prolines, and GST 4 (which included all isolates from 2018) had 23 prolines. GST 2 (including isolates from 2014 to 2016) and GST 3 (including isolates from 2016 to 2018) displayed PPL lengths of 19–22 prolines and 21–23 prolines, respectively. Each type had proline that exclusively included in the GST, giving more importance to the relation of the type and PPL length.

## 4. Discussion

Host-adapted *S. enterica* serovars have reduced virulence genes with increased pseudogenization, which is related to the elimination of some metabolic pathways for intestinal colonization of the bacteria [26]. *S.* Gallinarum has a restricted range of hosts that is primarily limited to poultry, and it shows significantly more degraded virulence genes and increased pseudogenes when compared with other *S*. *enterica* serovars [26,27]. Furthermore, the bacteria present few biochemical differences among strains [28], and field isolates show high levels of genetic similarity [29]. Despite the loss of virulence genes, increase of pseudogenes and homogeneous characteristics of *S.* Gallinarum, and attempts at eradication (e.g., vaccinations and antimicrobial treatments), FT is still prevalent among commercial poultry flocks from Korea, implying continuous virulence or a genetic change in the bacteria.

When the distribution of 20 virulence genes from the *S*. Gallinarum isolates was analyzed, all isolates displayed identical results. The *pefA* and *cdtB* virulence genes were absent from all 116 isolates. The *pefA* gene was not present because it is related to fimbriae, and *S.* Gallinarum is immotile [8]. The *cdtB* gene is primarily found in *Salmonella* Typhi [21]. Although there were some studies that revealed *cdtB* from other *Salmonella* serovars [30,31], we could not discover the gene in this study. Therefore, an increase in virulence of *S.* Gallinarum isolates could not be confirmed simply with or without virulence genes.

Kim et al. [12] reported that *S*. Gallinarum isolates from Korea display a longer PPL length than the other *S*. Gallinarum strains deposited in GenBank [11]; PPL length could be an indicator of the evolution of *S*. Gallinarum pathogenicity in Korea, although profound in vivo study is needed to establish the actual relationship between PPL length and virulence of strains. SpvB is an ADP-ribosyl transferase and destabilizes the cytoskeleton of the host cell [10]. PPL connects the N- and C-terminal domains of SpvB, and PPL length affects the translocation of SpvB to cytosol, with a longer PPL giving higher virulence activity to SpvB [12]. Kwon and Cho [11] reported that *S*. Gallinarum 9R, the live vaccine strain, had a PPL length of 9 prolines, while virulent *S*. Gallinarum isolates from Korea had PPLs of 17 prolines. Kim et al. (2019) [12] also reported that the most prevalent PPL length of *S*. Gallinarum isolated in 2016 was 15 prolines, while the longest PPL length in our study was 23 prolines in the isolate from 2016. However, in distribution of PPL length against *S*. Gallinarum isolates from 2014 to 2018 tested in this study, the most prevalent PPL length was 22 prolines, and the longest PPL length was 24 prolines in the isolate from 2018. Interestingly, a PPL length of 19 prolines, which was the shortest, was found only in isolates from 2014 and 2015. This study indicates that PPLs tend to lengthen over time in Korea, and it implies that the pathogenic potency of the field-isolated bacteria in Korea is increasing.

CRISPR typing is a powerful and efficient subtyping method in highly homogenous strains [18,19,24]. The 116 *S*. Gallinarum isolates tested in this study were divided into five distinct types with obvious spacer differences. Furthermore, there was significant GST variation from year to year, even though all *S*. Gallinarum isolates were isolated within a relatively short period of time. This result implies that *S*. Gallinarum was affected by various environmental factors [19,32] and that there was genetic transition of *S*. Gallinarum in poultry farms of Korea, accompanying the change of spacer distribution.

Specific GSTs were also associated with certain ranges of PPL lengths, with the exception of GST 5. This study showed that the transition of GST was associated with PPL length extension over time. GST 5 was the most prevalent type and had a wide range of PPL lengths, and the longest PPLs of 24 prolines appeared in isolates included in GST 5, so careful attention is needed on the type. Moreover, all three isolates included in GST 4 first emerged in 2018 and displayed PPLs of 23 prolines, so the novel type may cause more fatal disease than other types of bacteria. This result indicates that comprehensive surveillance of changing GST and PPL length are necessary in the monitoring of *S*. Gallinarum isolates for dissemination and that *S*. Gallinarum isolates with higher pathogenicity may appear in Korea in the future.

## 5. Conclusions

In this study, we investigated virulence and genetic variation of field *S*. Gallinarum isolates from Korea between 2014 and 2018. Although there was limitation to analyze virulence change through gene detection, the tendency of increasing virulence of the bacteria was identified through PPL length of SpvB, which is one of the pathogenicity decision factors. Especially, more pathogenic *S*. Gallinarum isolates could appear in the future if the current trend continues. In addition, GST presented obvious transition between years, which was correlated with virulence change. These variations made FT still prevalent in Korea despite many efforts to eradicate the disease. Therefore, comprehensive surveillance and monitoring of the molecular and virulence variation of *S*. Gallinarum is important for FT control and prevention.

## Figures and Tables

**Figure 1 animals-10-02346-f001:**
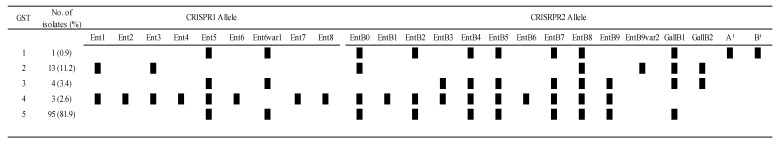
Spacer content of CRISPR alleles and grouping of 116 *Salmonella* Gallinarum isolates into Gallinarum Sequence Types (GSTs). Repeats are not included and only spacers are represented. ^1^ Spacers newly discovered in this study.

**Figure 2 animals-10-02346-f002:**
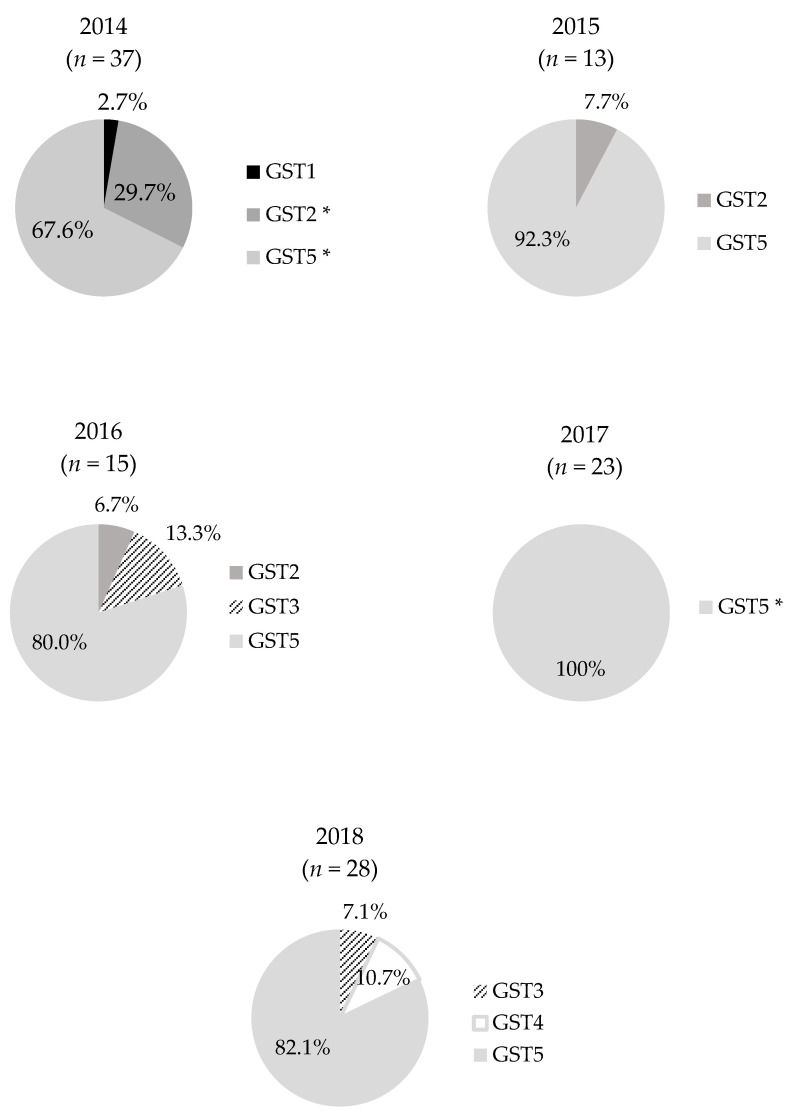
Pie charts of GSTs of 116 *Salmonella* Gallinarum isolates. Distribution of GSTs and their ratio of each year is presented on pie chart. Each color shows different GSTs as in the legend on the right side. * *p* < 0.05.

**Figure 3 animals-10-02346-f003:**
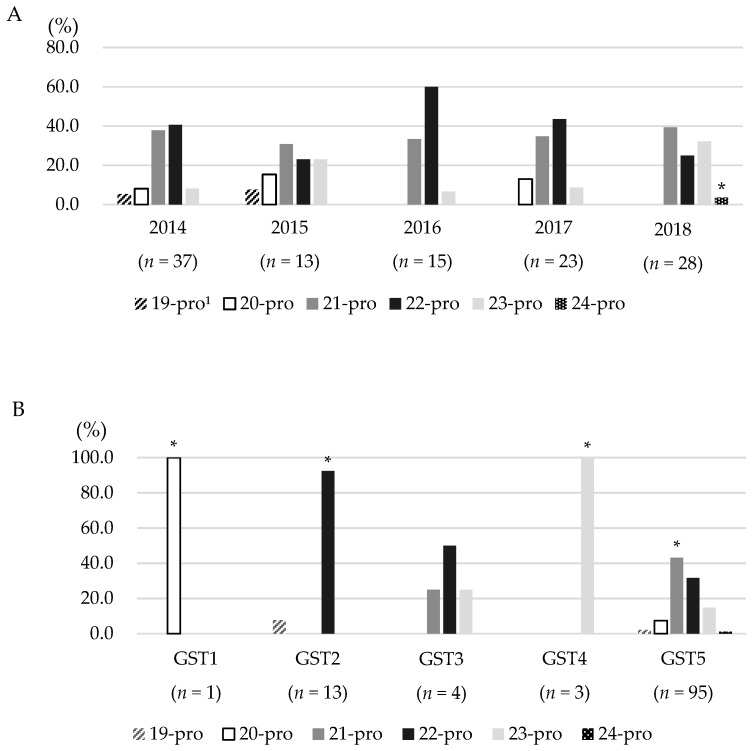
Bar graphs of polyproline linker (PPL) lengths percentage of *Salmonella* Gallinarum isolates. The graph represents PPL length variation from 2014 to 2018 (**A**) and according to GST (**B**). ^1^ The number of prolines in PPL region of SpvB. * *p* < 0.05.

**Table 1 animals-10-02346-t001:** Primers used in this study.

Genes	Function	Primer Sequence (5′→3′)	Size (bp)	References
*spiA*	Survival within macrophage	F: CCAGGGGTCGTTAGTGTATTGCGTGAGATG R: CGCGTAACAAAGAACCCGTAGTGATGGATT	550	[21]
*cdtB*	Host recognition /invasion	F: ACAACTGTCGCATCTCGCCCCGTCATT R: CAATTTGCGTGGGTTCTGTAGGTGCGAGT	268	[21]
*msgA*	Survival within macrophage	F: GCCAGGCGCACGCGAAATCATCC R: GCGACCAGCCACATATCAGCCTCTTCAAAC	189	[21]
*invA*	Host recognition /invasion	F: CTGGCGGTGGGTTTTGTTGTCTTCTCTATT R: AGTTTCTCCCCCTCTTCATGCGTTACCC	1070	[21]
*prgH*	Host recognition /invasion	F: GCCCGAGCAGCCTGAGAAGTTAGAAA R: GAAATGAGCGCCCCTTGAGCCAGTC	756	[21]
*spaN*	Entry into nonphagocytic cells	F: AAAAGCCGTGGAATCCGTTAGTGAAGT R: CAGCGCTGGGGATTACCGTTTTG	504	[21]
*orgA*	Host recognition /invasion	F: TTTTTGGCAATGCATCAGGGAACA R: GGCGAAAGCGGGGACGGTATT	255	[21]
*tolC*	Host recognition /invasion	F: TACCCAGGCGCAAAAAGAGGCTATC R: CCGCGTTATCCAGGTTGTTGC	161	[21]
*sitC*	Iron acquisition	F: CAGTATATGCTCAACGCGATGTGGGTCTCC R: CGGGGCGAAAATAAAGGCTGTGATGAAC	768	[21]
*lpfC*	Host recognition /invasion	F: GCCCCGCCTGAAGCCTGTGTTGC R: AGGTCGCCGCTGTTTGAGGTTGGATA	641	[21]
*sifA*	Filament structure formation	F: TTTGCCGAACGCGCCCCCACACG R: GTTGCCTTTTCTTGCGCTTTCCACCCATCT	449	[21]
*sopB*	Host recognition /invasion	F: CGGACCGGCCAGCAACAAAACAAGAAGAAG R: TAGTGATGCCCGTTATGCGTGAGTGTATT	220	[21]
*iroN*	Iron acquisition	F: ACTGGCACGGCTCGCTGTCGCTCTAT R: CGCTTTACCGCCGTTCTGCCACTGC	1205	[21]
*pagC*	Survival within macrophage	F: CGCCTTTTCCGTGGGGTATGC R: GAAGCCGTTTATTTTTGTAGAGGAGATGTT	454	[21]
*stn*	Enterotoxin	F: ATTGAGCGCTTTAATCTCCT R: GCTGTTGAATCTGTACCTGA	543	[22]
*sopE*	Colonization /host invasion	F: GGTAGGGCAGTATTAACCAG R: TTTATCTCCCTAGGTAGCCC	254	[22]
*pefA*	Host recognition /invasion	F: GCCAAAGTACTGGTTGAAAG R: TATTTGTAAGCCACTGCGAA	185	[22]
*sipB*	Entry into nonphagocytic cells	F: GGACGCCGCCCGGGAAAAACTCTC R: ACACTCCCGTCGCCGCCTTCACAA	875	[21]
*sefC*	Host recognition /invasion	F: GGCAGGTCCAAAACTATACA R: GCGATAACGAAACACCATTT	609	[22]
*spvB*	Growth within host /ADP ribose transferase activity	F: ATGTTGATACTAAATGGTTTTTCATC R: CTATGAGTTGAGTACCCTCATG	1776	[12]
PPL ^1^		F: GCACTTTTGAACAGGCCGTAG R: AGCTAGGCCGCTCATACCAC		[12]
CRISPR1 ^1^		F: GTTGGTAAAAGAGCTGGCGA R: GATGGACTGCGTTTGGTTTC	Varies	[19]
CRISPR2 ^1^		F: CAATACCCTGATCCTTAACG R: ATTGTTGCGATTATGTTGGT	Varies	[19]

^1^ Primers used for sequencing.

**Table 2 animals-10-02346-t002:** Presence of virulence genes in 116 *Salmonella* Gallinarum isolates.

Genes	2014 (*n* = 37)	2015 (*n* = 13)	2016 (*n* = 15)	2017 (*n* = 23)	2018 (*n* = 28)
*spiA*	● ^1^	●	●	●	●
*pagC*	●	●	●	●	●
*cdtB*	○ ^2^	○	○	○	○
*msgA*	●	●	●	●	●
*invA*	●	●	●	●	●
*sipB*	●	●	●	●	●
*prgH*	●	●	●	●	●
*spaN*	●	●	●	●	●
*orgA*	●	●	●	●	●
*tolC*	●	●	●	●	●
*iroN*	●	●	●	●	●
*sitC*	●	●	●	●	●
*lpfC*	●	●	●	●	●
*sifA*	●	●	●	●	●
*sopB*	●	●	●	●	●
*sopE*	●	●	●	●	●
*pefA*	○	○	○	○	○
*sefC*	●	●	●	●	●
*stn*	●	●	●	●	●
*spvB*	●	●	●	●	●

^1^ Solid circle (●) means that the gene was presented in all *S*. Gallinarum isolates tested; ^2^ Open circle (○) means that the gene was presented in none of the *S*. Gallinarum isolates tested.

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
