# Peer review of "Virulence Variation of Salmonella Gallinarum Isolates through SpvB by CRISPR Sequence Subtyping, 2014 to 2018"

_animals, 2020, doi:10.3390/ani10122346_

Round 1

Reviewer 1 Report

The authors of the manuscript “Virulence variation of Salmonella Gallinarum isolates through SpvB by CRISPR sequence subtyping, 2014 to 2018” addressed most of my comments. However, the text still can be improved grammatically and stylistically.

I would like to point out that some improvements still could be made in the text:

  1. The title can be improved: “Virulence variation of Salmonella Gallinarum isolates based on the length of SpvB and CRISPR sequence subtyping, 2014 to 2018”
  2. Line 16: change to “for monitoring of S. gallinarum dissemination ”
  3. Line 20: “the disease, the disease” change to “the disease, the malady”
  4. Line 25 : factors
  5. Line 37: the infection of
  6. Line 38-41: change to “the disease has several transmission routes: though food, environment or contaminated eggs, causing significant financial losses for farming industries.”
  7. Line 69 : ‘ highly homogenous strains ’
  8. Line 69 : change to “methods are less efficient at stratifying the strain into several subgroups”’
  9. Lines 87-96 : Change C to 0C
  10. Table 1 : Change Variable to “varies”
  11. Line 111: the link worked in the rebuttal text, but not in the main text
  12. Line 138: “nominated” should be changed to “named” or “labeled”. The sentence still has to mention Ent, Gall, A and B labels.
  13. Line 146 : change “expressed” to “labeled as” or “named as”
  14. Line 183: prolines
  15. Line 200: change to “in this study”
  16. Line 205-206: change to “to establish the actual relationship”
  17. Line 240: change to “could appear in the future if the current trend continues.”
  18. Line 244: change “to deal with FT.” to “for FT control and prevention.”

Author Response

Dear reviewer 1.

Thank you for your detailed review of my article. I gladly corrected all of your recommendations except the article’s title.

I thought ‘the length of SpvB’ is a little bit different from my intention, because PPL length is missing in the expression. However, SpvB is lager meaning that includes PPL and covers my intention, so I decided not to fix the article’s title.

All other points the reviewer checked out were reasonable and grammatically correct, so I followed the recommendation.

Kind regards,

Ms. Koeun Kim.

Reviewer 2 Report

the authors reliably referred to the reviews and corrected the flaws

Author Response

Dear reviewer 2.

Thank you for your review of my article.

The reviewer pointed out that introduction should be more extensive and attractive. However, the introduction includes enough background information that is needed to understand the whole article. The disease occurs in limited country, so it may be unattractive to countries where FT is already eradicated, so I emphasized its damage in Korea. I hope this will satisfy the reviewer.

Kind regards,

Ms. Koeun Kim.

This manuscript is a resubmission of an earlier submission. The following is a list of the peer review reports and author responses from that submission.

Round 1

Reviewer 1 Report

Fowl typhoid caused by Salmonella Gallinarum is a significant problem in numbers of countries, although the disease has been eradicated in others through strict measures including vaccination and slaughter programs. The prevalence of this disease South Korea is attributed to a combination of factors, including an ineffective vaccine, a more susceptible breed of chicken and the prevalence of a mite that helps transmit S. Gallimarum. In this study the authors continue the characterization of 116 of the 130 S. Gallimarum strains that were isolated from 8 different provinces in South Korea. They determined the strains to be highly similar with regard the 20 virulence genes whose presence they tested via PCR, but showed more diversity with regard the length of the polyproline linker (PPL) of the spvB gene, which has been implicated in virulence, and also the content of their CRISPR arrays.

The overall impact of the research is lowered by the presentation of the results and methods. The way the studies were performed and described means the longer term context to others working on this or related organisms is extremely limited, there are no control strains employed. The authors do not even put their findings into geographic context (although they would have that data and it would be relevant to growers) nor with their previous studies on the antibiotic resistance profiles of these strains, and the presence/absence of genes associated with drug resistance that was presented for the 39 strains that had had multi-drug resistant profiles.

At a time when the cost of whole genome sequencing has reduced significantly, it might be as quick/economic to sequence these strains versus using time intensive/error prone PCR. Even if whole genome sequencing was only done on a few isolates, the global profile of these strains would be more informative both to characterizing the genetic differences which may be responsible for the virulence of S. Gallinarum and to a wider audience.

  • Methods are inadequately described, other than referring to three different manuscripts and provision of primer sequences, the description of the PCR of virulence genes is inadequate, making it impossible to replicate the experiments
    • g., which polymerase was used, other reaction component and conditions?
    • Which company synthesized the primers, what was their purity and concentration
    • Was multi-plex PCR employed (as used in one of the citations provided) or were single reactions for each gene used
      • If the latter why not sequence the entire genome of these isolates, the information that would be so much richer as whole genome comparisons could be done.
    • Were any controls included?
      • e., is the presence/absence of any virulence genes purely from a false positive or negative PCR result?
    • How were PCR product sizes determined (i.e., gel conditions/ molecular weight marker used)
  • Similarly, the description of the analyses of the CRISPR arrays is inadequate
    • The primers for the CRISPR arrays were described as being validated (Lines 86-89) by Blast but which genome or sequence were the primer sequences validated against?
      • Why was the validation conducted/needed when (a) it was not done for the other primers utilized in the study and (b) the primers used were already published?
    • The URL provided (Lines 93/94) – gave me the error “could not find tool” on three different browsers, I did find a link (https://research.pasteur.fr/en/tool/salmonella-crispr-typing/) that then gave me a link to software available through Github, presumably this is what the authors employed?
      • Unfortunately, the original manuscript cited by Fabre et al. has a reference who link is also dead: https://journals.plos.org/plosone/article?id=10.1371/journal.pone.0036995 provides a link for a tool that is no longer available: http://mlst.ucc.ie  - so it is critical that these details are accurate in this manuscript
    • Were the same primers used for sequencing of the CRISPR arrays as those used for the PCR, were any nested primers required?

Results

  • See note above in methods re was multi-plex PCR conducted or not? A representative PCR figure, with appropriate controls, would be helpful
  • Figure 1 - The GST typing is confusing – what is the “Ent” referring to in the name for the spacers in Fig 1?
    • Enterica?
    • Do the “Ent” numbers in some way correspond to something in the Fabre et al manuscript?
    • Do they refer to position in the array and/or a sequence?
    • There seems to be great variability in the total numbers of spacers in both loci in different strains? Presumably then the authors obtained PCR products of different lengths to reflect this?
    • It would be helpful to have the CRISPR sequences of each strain included (possibly in Supplementary data)
    • It is not clear for the spacers “A” and “B” in CRISPR locus II, if the entire spacer is a new insertion in those strains and the sequence different to any previously reported (if so can the authors identify a phage/entity from which the spacers may have orginiated), or do “A and B” spacers represent SNP variations?
    • And where in the CRISPR loci are the new spacers located – on the 5’ end where you would expect to observe a new acquisition?
    • This is important to clarify since numbers of Salmonella have been shown to not be acquiring new CRISPR spacers, but rather show variability in these regions via “microevolution” which includes mutations etc occurring to the existing arrays.
  • The authors should address whether whole genome sequencing of these strains, or even representatives (eg with major variations in multi-drug resistance profiles or from different locales) would give a clearer picture as to the changes relative to a reference strain that would clearly delineate all changes, not just a subset of virulence genes and CRISPR loci.

Reviewer 2 Report

Salmonella enterica serovar Gallinarum can cause fowl typhoid leading to significant financial losses for the poultry industry especially in some Asian countries.  The authors Kim et al. of the manuscript “Virulence variation of Salmonella Gallinarum isolates based on CRISPR sequence, 2014 to 2018” divided 116 isolates of bacteria into 5 sequence types based on polymorphism in CRISPR regions in the genome of Salmonella enterica serovar Gallinarum. CRISPR types also contained different frequency of polyproline linker (PPL) length between N- and C-terminal domains of the virulence factor SpvB. The analysis showed a tendency towards increase in PPL length from 2014 to 2018, which is indicative of ever growing pathogenicity of the bacteria.

The manuscript is well-written and well-structured; I would like to propose only some minor changes.

  1. The title of the manuscript is a bit confusing:
    - the title should reflect the main finding about the length of PPL region of SpvB
    - CRISPR sequence screen is not a novel method, therefore it should not be in the title, the title should reflect some degree of novelty
    - “CRISPR sequence” in the title can be misleading to those expecting to see more information about CRISPR gene editing
  2. In the abstract please mention that Salmonella is a Gram-negative bacteria.
    The abstract should state clearly that CRISPR sequence analysis was used for bacteria subtyping and the length of PPL region of SpvB was tested to assess the pathogenicity
  3. "but still presents in Korea" – should it be just “present”?
  4. "Therefore, surveillance of genetic change and virulence increase is necessary" - surveillance of genetic change associated with virulence increase – would sound better in my opinion.
  5. In the introduction the benefit of the described CRISPR typing/PCR/sequencing method has to be emphasized and explained why it is still better than conventional methods such as serotyping.
  6. "FT still presents in Korea." - FT appears in the abstract and the abbreviation is not introduced earlier (but is explained in the introduction later)
  7. "polyproline linker (PPL) length of the SpvB" - SpvB is not introduced in the text of the abstract (is it a virulence factor?). PPL - that is the position of the linker relative to the SpvB protein length? Maybe it has to be already mentioned in the abstract.
  8. "red poultry mites, which is a significant risk factor" - which are a significant ...?
  9. Table 2 - not sure if it is needed? the conclusion for this table is pretty solid from the text.
  10. lines 106-108 - hard to understand, please consider re-writing.
  11. Fig1 - maybe somewhere in the text/legend it should be mentioned that Ent and Gall are the spacer names?
  12. "CRISPR1 and CRISPR2 loci showed 3 and 5 CRISPR allelic types, respectively." - where can I see 3 and 5 allelic types? Unclear sentence, please re-write.
  13. "GST 1 had 2 novel spacers which were not present in “Institute Pasteur CRISPR database for Salmonella” 113 program database, so they are expressed as A and B in Figure 1."
    - I can only see the GST1 with spacers A and B
  14. Figure 2 – the pie chart from 2017 shows 100%, please remove the white line (it is confusing as it can be interpreted as a tiny portion of the pie chart)
  15. Figure 2 - GST3 in 2016 and 2018 doesn't seem to have the same color coding. It is hard to see the difference between GST3 and 4 in the pie chart from 2018. Please consider a different color scheme.
  16. Figure 2 - "presented on the legend of right side" - in the legend on the right side
  17. Discussion about physiological relevance of CRISPR repeats is needed.
  18. Fig3. Perhaps, a graph with a time line vs GST and proline rich prevalence would be more descriptive (GST prevalence by year was already in the Fig2, but it is hard to remember at times when looking at the Fig3B what year was associated with each GST).
  19. Materials and methods - the galaxy link doesn't work

Reviewer 3 Report

This manuscript reports a study on molecular evolution analysis of Salmonella gallinarum field isolates  with regard to the genetic variation of virulence indicated by CRISPR-based typing and PPL lengths of  SpvB. Studies were performed since 2014  on 20 representative virulence genes.

Given this report and the literature, I believe this to be a interesting manuscript and important to the agriculture sector in Korea (place of the study) due to its observable prevalence in poultry sector in Asia. Based on my experience in molecular biology research, the methodology is proper and up-to-date.

Overall the article is written well, but there are some minor errors that need to be resolved.

The most important disadvantages of the article are listed below

Major flaws:

-The introduction is modest and unsatisfactory. It should introduce the reader more to the topic of salmonellosis in poultry . The introduction could be more extensive, in particular with regard to the molecular research carried out.

-Statistics are completely missing in the reviewed article. I am convinced that some data could be analyzed statistically, so that you could obtain additional, probably interesting and valid results. Authors should consider statistical analysis of PPL lengths percentage of Salmonella gallinarum isolates and distribution of GSTs and their ratio per each year. Thanks to this, the authors could observe whether there are statistically significant differences between the examined factors

-I absolutely agree that the conducted research may indicate virulence or pathogenicity of Salmonella spp. Nevertheless, I miss the discussion that these results should be confirmed in vivo tests. The authors were able to collect clinical data on salmonellosis in flocks tested. This would allow the symptoms and their intensity to be related to the results described. If the authors do not have such data, the discussion should engage in a polemic about linking research results with studies on living organisms and then draw more serious conclusions. Without this, we are not able to say whether the length of the PPL does not affect disease processes at some point (i.e pathogenicity may increase up to 22-pro but with 23-pro and 24-pro the symptoms intensity increase or significant differences in mortality might not be observe). Other useful data could be flocks mortality or disease duration, which would also show the actual relationship between pathogenicity and genetic variation. Authors should emphasized this in discussion, that on the basis of these results, thorough clinical trials should be conducted.

Minor flaws:

In the attachment I am sending an article with minor errors that require improvement / reflection.

To sum up, in my opinion the article “Virulence variation of Salmonella Gallinarum isolates based on CRISPR sequence, 2014 to 2018” is potentially acceptable for publication in “Animals”, once the Authors carried out some essential revisions
